# Predictive Biomarkers of COVID-19 Severity in SARS-CoV-2 Infected Patients with Obesity and Metabolic Syndrome

**DOI:** 10.3390/jpm11030227

**Published:** 2021-03-22

**Authors:** Carles Perpiñan, Laia Bertran, Ximena Terra, Carmen Aguilar, Miguel Lopez-Dupla, Ajla Alibalic, David Riesco, Javier Camaron, Francesco Perrone, Anna Rull, Laia Reverté, Elena Yeregui, Anna Marti, Francesc Vidal, Teresa Auguet

**Affiliations:** 1Institut Català de Salut (ICS), 43202 Reus, Spain; cperpinan.tgn.ics@gencat.cat; 2Grup de Recerca GEMMAIR (AGAUR)-Medicina Aplicada, Departament de Medicina i Cirurgia, Universitat Rovira i Virgili (URV), Institut d’Investigació Sanitària Pere Virgili (IISPV), 43007 Tarragona, Spain; laia.bertran@urv.cat (L.B.); caguilar.hj23.ics@gencat.cat (C.A.); driesco.hj23.ics@gencat.cat (D.R.); 3Grup de Recerca MoBioFood, Departament de Bioquímica i Biotecnologia, Universitat Rovira i Virgili (URV), Institut d’Investigació Sanitària Pere Virgili (IISPV), 43007 Tarragona, Spain; ximena.terra@urv.cat; 4Servei Medicina Interna, Hospital Universitari Joan XXIII Tarragona, 43005 Tarragona, Spain; jmlopezdupla.hj23.ics@gencat.cat (M.L.-D.); aalivalic.hj23.ics@gencat.cat (A.A.); javiercamaron93@gmail.com (J.C.); fgperrone.hj23.ics@gencat.cat (F.P.); fvidalmarsal.hj23.ics@gencat.cat (F.V.); 5Grup de Recerca INIM, Departament de Medicina i Cirurgia, Universitat Rovira i Virgili (URV), Institut d’Investigació Sanitària Pere Virgili (IISPV), 43005 Tarragona, Spain; anna.rull@iispv.cat (A.R.); laia.reverte@iispv.cat (L.R.); eyeregui.hj23.ics@gencat.cat (E.Y.); anna.marti@iispv.cat (A.M.)

**Keywords:** COVID-19, SARS-CoV-2, obesity, metabolic syndrome, pneumonia, prediction, personalized therapy

## Abstract

In SARS-CoV-2-infected patients, obesity, hypertension, and diabetes are dangerous factors that may result in death. Priority in detection and specific therapies for these patients are necessary. We wanted to investigate the impact of obesity and metabolic syndrome (MS) on the clinical course of COVID-19 and whether prognostic biomarkers described are useful to predict the evolution of COVID-19 in patients with obesity or MS. This prospective cohort study included 303 patients hospitalized for COVID-19. Participants were first classified according to the presence of obesity; then, they were classified according to the presence of MS. Clinical, radiologic, and analytical parameters were collected. We reported that patients with obesity presented moderate COVID-19 symptoms and pneumonia, bilateral pulmonary infiltrates, and needed tocilizumab more frequently. Meanwhile, patients with MS presented severe pneumonia and respiratory failure more frequently, they have a higher mortality rate, and they also showed higher creatinine and troponin levels. The main findings of this study are that IL-6 is a potential predictor of COVID-19 severity in patients with obesity, while troponin and LDH can be used as predictive biomarkers of COVID-19 severity in MS patients. Therefore, treatment for COVID-19 in patients with obesity or MS should probably be intensified and personalized.

## 1. Introduction

Coronavirus disease 2019 (COVID-19), a disease caused by severe acute respiratory syndrome coronavirus 2 (SARS-CoV-2), has emerged as a rapidly spreading communicable disease, affecting more than 190 countries across the globe [1,2]. The lack of vaccines until now or effective treatments for the disease and the low immunity of the population to SARS-CoV-2 infection have led to the overgrowth of the pandemic, which has caused a social and economic crisis around the world [3]. The disease is primarily spread through large respiratory droplets, although the possibility of other routes of transmission cannot be ruled out, as the virus has been found in the stool and urine of affected individuals [4].

COVID-19 severity varies from mild self-limiting flu-like illness to fulminant pneumonia, respiratory failure, and death. Factors associated with severity and worse prognosis of COVID-19 are age, the presence of chronic diseases, such as type 2 diabetes mellitus (T2DM) or arterial hypertension, and those with a compromised immune system [5]. In a published series, obesity was also associated with severe disease, especially in people under 65 years [6,7].

Obesity is a complex chronic metabolic and multifactorial disease that, associated with a chronic inflammatory state, has an essential role in the development of T2DM, dyslipidemia, hypertension, cardiovascular disease (CVD), or cancer [8]. Obesity is associated with a worse immune response and a poor prognosis for respiratory infections, as evidenced during the influenza A (H1N1) epidemic in 2009 [9,10]. In this sense, people with obesity or T2DM are at risk of pneumonia infection, which can be reduced with good glycemic control [11,12]. Some analyses of fatal cases of coronavirus-related pneumonia have shown that comorbidities, such as obesity, hypertension, or T2DM (components of metabolic syndrome (MS)), are dangerous factors that may result in death [1,13,14].

The damage inflicted by COVID-19 is primarily mediated through an excessively exuberant host response to the virus, not directly by the virus itself, which consists of excessive and prolonged inflammation, culminating in a cytokine storm and disseminated coagulopathy, injuring the lungs and blood vessels [5].

Subjects with obesity and T2DM have disrupted innate and adaptive immune responses [15]. These patients have a chronic low-grade inflammation state due to metabolic dysfunction. Chronic inflammation causes decreased macrophage activity and increased production of proinflammatory cytokines, leading to enhanced susceptibility and a delay in resolution of the viral infection [14].

Moreover, it has been suggested that metabolic dysregulation is also the underlying cause of fibrosis, such idiopathic pulmonary fibrosis (IPF). Zank et al. suggested that metabolic changes in IPF have a potential impact on lung cell function, differentiation, and activation of fibrotic responses [16]. In this regard, adipocytokines, molecules secreted by adipocytes, macrophages, and other cell types dysregulated in obesity and MS are involved in the regulation and mediation of inflammatory response of IPF [17,18].

Obesity is a known risk factor for a worse prognosis in COVID-19 pneumonia. However, the relationship between MS and the disease evolution is unclear. In this sense, in the present study, we aimed to investigate the association of MS with severity criteria for COVID-19 pneumonia. Moreover, especially, we sought to study whether prognostic biomarkers described in patients with COVID-19 were capable of predicting the evolution of disease according to the World Health Organization (WHO) classification of severe pneumonia by COVID-19, in patients with this infection who have obesity or MS [19].

## 2. Materials and Methods

### 2.1. Subjects

This is a single center cohort study. A total of 303 patients consecutively admitted to Internal Medicine Department in University Hospital of Tarragona Joan XXIII for SARS-CoV-2 were enrolled between 2 February and 26 September 2020. All patients were diagnosed with COVID-19 pneumonia according to WHO guidance [19]. Throat swab samples were obtained from all patients at admission and tested for COVID-19 using real-time reverse transcriptase–polymerase chain reaction (PCR) assays. The study was approved by the institutional review board (Institut d’Investigació Sanitària Pere Virgili CEIm; 079/2020), and all participants gave informed consent.

### 2.2. Data Collection 

A trained team of physicians collected epidemiological data, past medical history, treatments, clinical data, and outcomes for all patients. Patient confidentiality was protected by assigning an anonymous identification code, and the electronic data were stored in a password-protected computer.

The following patient characteristics were analyzed: sex, age, body mass index (BMI), background, comorbidities, radiological findings, clinical characteristics, and biochemical parameters. On the one hand, the 303 study participants were first classified according to their BMI (the severity of the disease in these patients at the time of admission did not allow obtaining weight and height adequately only from 243 subjects): patients without obesity (BMI < 30 kg/m^2^; *n* = 176) and with obesity (BMI ≥ 30 kg/m^2^; *n* = 67). On the other hand, the whole cohort of patients were classified depending on the presence of MS (MS, *n* = 48; non-MS, *n* = 255) according to Alberti et al. criteria [20]. The severity of the disease will be classified according to the WHO eight-point classification of severe pneumonia by COVID-19 [19], namely: (0) no clinical or virological evidence of COVID-19 infection; (1) infected without limitations; (2) limitation of activity; (3) hospitalized without oxygen therapy; (4) oxygen by mask or nasal; (5) non-invasive ventilation (NIV) (continuous positive airway pressure (CPAP) or positive bipressure in the airways (BiPAP)) or high-flow oxygen (HFO); (6) intubation with mechanical ventilation (MV), mask with reservoir; (7) MV or extracorporeal membrane oxygenation (ECMO), support with vasopressors, or dialysis/renal replacement therapy; (8) death.

We collected all the data including clinical, laboratory parameters, the treatment established for COVID-19, comorbidities and background, chest computed tomography (CT), and anthropometrical evaluation.

### 2.3. Biochemical Analysis

At the time of admission, all patients were examined in the laboratory, including blood routine, blood biochemistry, coagulation function, infection-related biomarkers, and co-infection. Blood samples were obtained from all participants. Biochemical parameters were analyzed using a conventional automated analyzer after 12 h of fasting.

### 2.4. Statistical Analysis

The data were analyzed using the SPSS/PC+ for Windows statistical package (version 23.0; SPSS, Chicago, IL, USA). All results were expressed as mean (standard deviation, SD) or median (interquartile range, IQ) for continuous variables and as frequency (percentage) for categorical variables. Continuous variables were compared between groups with the Student *t*-test or Mann–Whitney U-test according to their distribution. Categorical variables were compared with the Chi squared or Fisher’s exact test. The strength of the association between variables was calculated using the Spearman ρ correlation test (nonparametric variables). *p* values < 0.05 were statistically significant. To calculate whether certain variables could predict the evolution of the disease (according to the WHO classification), we used the Classification and Regression Tree (CRT) method. Variables included to generate the regression tree were age, gender, BMI, systolic blood pressure (SBP), diastolic blood pressure (DBP), white blood cell count, lymphocyte count, D-dimer, erythrocyte sedimentation rate (ESR), interleukin (IL)-6, ferritin, fasting glucose, creatinine, aspartate aminotransferase (AST), alanine aminotransferase (ALT), gamma-glutamyl transferase (GGT), alkaline phosphatase (AP), lactate dehydrogenase (LDH), and troponin. CRT analysis splits the data into segments that are as homogeneous as possible with respect to the dependent variable.

## 3. Results

### 3.1. Baseline Characteristics of Participants

The main characteristics of the whole study cohort are described below. The median age was 62 years (IQR 45–74), BMI 26.55 kg/m^2^ (IQR 23.92–30.37): (BMI < 30 kg/m^2^: 72.24%, BMI ≥ 30 kg/m^2^: 27.6%), and 52.5% of patients were male. Background: 42.2% of patients exercised regularly, 77.8% were non-smokers, 9.3% were active smokers and 12.9% former smokers, 17.2% had T2DM, 33% had dyslipidemia, 40.6% had arterial hypertension, 52% previous CVD, 9.6% had some previous respiratory disease, and 8.9% had a history of cancer. 

Regarding clinical characteristics, 5.3% of patients were asymptomatic from the respiratory point of view (they had been admitted for other unrelated illnesses and were diagnosed through a pre-hospitalization screening program), 24.4% had mild symptoms, 50.5% had moderate symptoms, and 19.8% had critical symptoms; 43.6% of patients presented respiratory failure and 16.8% needed admission to the ICU. The mortality rate rose up to 11.9%. 

Regarding radiologic characteristics, bilateral interstitial pattern was observed in chest x-ray or CT in 59.7% of cases, and pleural effusion in only 1%. 

The treatment established for COVID-19 was hydroxychloroquine in 42.9% of cases, azithromycin in 40.7%, lopinavir–ritonavir in 31%, tocilizumab in 4.3%, interferon in 4.6%, corticosteroids in 29%, and remdesivir in 15.8%. This treatment has varied throughout the period studied, according to the clinical guidelines update.

In the regard of oxygen therapy, 50.2% of patients needed oxygen by mask or nasal, 15.2% high-flow nasal cannulas (HFO), 2.3% NIV-CPAP, and 0.7% NIV-BiPAP; 14.2% of patients needed MV by intubation, and 8.3% by mask with reservoir; 11.6% of patients needed vasopressors with ECMO and 2.6% dialysis.

### 3.2. Evaluation of Clinical Outcome of COVID-19 Regarding the Presence of Obesity and Metabolic Syndrome

Obesity substudy: Data are summarized in Table 1. When we compared patients with obesity (BMI ≥ 30 kg/m^2^, 27.6% of the general cohort) or without obesity (BMI < 30 kg/m^2^, 72.24%), we observed they were comparable in terms of age and gender. However, patients with obesity did less physical exercise, had T2DM or hypertension more frequently, and, consequently, presented more MS. Patients with obesity often suffer from moderate COVID-19 symptoms and moderate pneumonia, while patients without obesity usually have mild symptoms. 

On the other hand, patients with obesity presented bilateral interstitial pattern in chest x-ray or CT more frequently than patents without obesity. However, we did not find any significant differences regarding ICU admission or the mortality rate. Additionally, patients with obesity received tocilizumab more frequently than non-obese patients. We also found significant differences regarding fasting glucose, total cholesterol, and LDH levels. We did not observe any significant differences regarding the following analytical values (leukocytes, lymphocytes, D-dimer, ESR, ferritin, C-reactive protein (CRP), IL-6, high-density lipoprotein-cholesterol (HDL-c), low-density lipoprotein-cholesterol (LDL-c), triglycerides, creatinine, ALT, AST, GGT, AP and troponin).

MS substudy: Data are shown in Table 2. Patients with MS (15.8% of the general cohort) were older, more frequently male, with higher BMI and higher SBP, did exercise in a lower percentage, and were former smokers more frequently than non-MS patients. MS subjects had previous CVD or respiratory illnesses more frequently than those who did not have MS. Regarding the prognosis of COVID-19, patients with MS presented more frequently moderate symptoms than patients without MS such as severe pneumonia or respiratory failure. The mortality rate was markedly increased in MS patients. With regard to biochemical parameters, MS patients showed higher levels of fasting glucose, creatinine, and troponin levels compared to non-MS patients. 

Finally, we analyzed whether sex or age can differentially influence the development of the disease in hospitalized patients with obesity and MS. In this sense, we did not find significant differences between sexes when we analyzed COVID-19 severity variables such as symptomatology, ICU admission, respiratory failure, the necessity of a specific treatment, or the necessity of oxygen therapy, among others. On the other hand, when we analyzed differences in patients with obesity and MS younger than 65 years old and older, we found that older patients have a higher risk of presenting respiratory failure (*p* = 0.024), needing MV by intubation (*p* = 0.049), or having a fatal outcome (*p* = 0.009).

### 3.3. Correlation between Clinical and Biochemical Variables with COVID-19 Severity

We also wanted to analyze the relationship between general, clinical, and biochemical variables with the severity of COVID-19 according to the WHO eight-point classification of severe pneumonia by COVID-19 [19]. We elaborated a correlation table with the significant (*p*-value < 0.05) statistical data obtained from the Spearman association test as shown Table 3.

### 3.4. Predictive Value of Clinical and Biochemical Variables of COVID-19 Outcome

Our last goal was to analyze whether some variables were capable of predicting the evolution of the disease, according to the WHO classification of severe pneumonia by COVID-19. The CRT method was used to obtain the best cut-off values of each variable to predict the outcome. First, we created a tree of the whole cohort of study, entering, on one hand, clinical variables (WHO classification: 4, 5, 6, 7, 8) and, on the other hand, the white blood cell count, lymphocyte count, D-dimer, ESR, IL-6, ferritin, fasting glucose, creatinine, AST, ALT, GGT, AP, LDH, and troponin levels as independent variables. The main results are shown in Figure 1. In Figure 1A, the prediction was that 100% of patients with IL-6 levels higher than 58.75 pg/mL would need oxygen by mask or nasal (WHO 4). Figure 1B shows that it was expected that 88.5% of patients with a CRP lower than 19.15 mg/dL will not need NIV (WHO 5). Figure 1C shows on the one hand, that 87.5% of patients with IL-6 levels lower than 88.30 pg/mL would not need MV by intubation (WHO 6); meanwhile, 76.9% of patients with an IL-6 higher than 88.30 pg/mL would need it. On the other hand, if we analyzed GGT from patients with IL-6 levels lower than 88.30 pg/mL, we observed that 92.9% of patients with a GGT lower than 98.50 U/L would not need MV by intubation. In Figure 1D, the classification tree analysis revealed that 90.4% of patients with LDH levels lower than 527.5 U/L would not need MV, nor support by vasopressors or dialysis (WHO 7). Finally, in Figure 1E, we can observe that 92.3% of patients with IL-6 levels lower than 62.75 pg/mL would not die by COVID-19.

Second, we created another decision tree with patients with obesity using CRT method. As shown in Figure 2, it was expected that 97.9% of patients with IL-6 levels lower than 51.75 pg/mL would not die by COVID-19. 

Finally, we created the last decision tree with MS cohort of patients, and we found significant classifications regarding WHO 4 (oxygen therapy) and WHO 6 (MV). As we can see in Figure 3A, the 92.6% of patients with troponin levels higher than 11.5 ng/L would need oxygen by mask or nasal. In Figure 3B, we can observe that 85.7% of patients with LDH levels lower than 311.5 U/L would not need MV, while 72.7% of patients with LDH levels higher than 311.5 U/L would need it.

## 4. Discussion

The novelty of this work lies in the fact that we aimed to study, in a well-characterized cohort of SARS-CoV-2-infected patients, the association between MS-related parameters and the clinical course of COVID-19. Moreover, we wondered whether any of the clinical and biochemical variables were capable of predicting the evolution of the disease, according to the WHO classification in the whole cohort, but especially in the population with obesity or MS.

First, we analyzed the impact of obesity on COVID-19 pneumonia evolution in our cohort. The main findings regarding obesity are that our patients with obesity presented moderate COVID-19 symptoms and bilateral interstitial patterns and received tocilizumab more frequently. A French study reported a significant association between the prevalence of obesity and severe COVID-19, including critical COVID-19, suggesting that obesity might be a risk factor for pejorative evolution of COVID-19, increasing the risk of ICU admission [21]. Additionally, mortality was higher among those who had obesity [22] or poor prognosis in this group of patients [5,23,24,25]. However, in our study, patients with obesity did not enter the ICU more frequently than patients without obesity, nor did they have higher mortality, perhaps because this cohort did not have severe obesity. Similarly, other authors, in a meta-analysis, concluded that obesity was associated with increased mortality only in studies with fewer chronic or critical patients [26].

To achieve the main objective of our study, we explored the influence of MS on the COVID-19 clinical course. Patients with MS were older and more frequently had previous comorbidities, such as CVD or respiratory illnesses. Moreover, they presented with a higher frequency of severe pneumonia or respiratory failure than patients without MS. Interestingly, the mortality rate was increased in MS patients, according to Cunningham et al., who described higher mortality among patients with obesity and hypertension [22], two MS components. Consistently, studies from Chinese cohorts of patients with COVID-19 have identified several risk factors for severe COVID-19, including obesity-related complications, such as T2DM and hypertension [27]. Consistent with our results, Marhl et al. reported that MS and T2DM are two relevant risk factors for COVID-19 [28].

Regarding pathophysiology, it has been suggested that people with obesity and MS display a higher risk of mortality and morbidity to COVID-19, especially due to exacerbated inflammatory status and hypercoagulation tendency [29]. All these observations call for increased vigilance, priority in detection and testing, and specific therapies for patients with obesity-MS and COVID-19 infection [5,11].

Additionally, other variables were related to poor prognosis. First, age correlated with some WHO severity criteria, including the need for oxygen therapy, NIV, and MV until death, as previously described. Age is the most common risk factor for COVID-19 severity, and older age has consistently been associated with higher mortality in patients with COVID-19 [30,31]. 

In contrast, the habitual practice of physical activities showed a negative correlation with WHO 4 (the need of oxygen by mask or nasal) and WHO 8 (death), which reinforces the fact that the practice of physical exercise seems to strengthen the immune system, suggesting a benefit in the COVID-19 clinical course [32].

Regarding comorbidities, such as T2DM, hypertension, CVD, respiratory illness, or oncological diseases, these were positively correlated with different WHO criteria of COVID-19 severity, according to a recent comprehensive systematic review and meta-analysis including 60 studies conducted in hospitals of 13 countries [26]. These findings reinforce the fact that chronic comorbidities are important risk factors for worse COVID-19 prognosis [33].

Although the use of specific pharmacologic treatments for COVID-19 has also been associated with different severity states of the disease, we cannot draw conclusions in this regard, since these treatments varied throughout the period studied, according to clinical guideline updates. Therefore, the lack of samples precluded such an analysis. However, it is worth noting that patients with obesity needed tocilizumab, an anti-IL-6 receptor antibody [34], more frequently than patients without obesity when PaO2/FiO2 worsened.

According to what was described in previous studies [35,36], in our cohort of patients, analytical parameters such as lymphocyte count, D-dimer, IL-6, CRP, GGT, LDH, and troponin had a positive association with worse prognosis (severity range between WHO 4 and 8).

The last objective of the present work was to assess a prediction of disease evolution through the CRT method. We found that the best predictor of necessity of oxygen by mask or nasal, the need of MV, or death was IL-6. Other studies have also described IL-6 as a predictor of mortality associated with COVID-19 [37,38] or of morbidity [39,40]. When we assessed the CRT method only in the patients with obesity, we also found that the best predictor of mortality was IL-6. Therefore, IL-6 is a biomarker to predict COVID-19 severity, and also fatal outcomes both in the general SARS-CoV-2 infected population and in COVID-19 patients with obesity. Additionally, we also assessed the CRT method in the MS cohort of study, and, as a novelty, we found that troponin is a good biomarker to predict the need of oxygen therapy; an LDH is a good predictor of MV or HFO necessity in COVID-19 patients. Determination of these predictive biomarkers of COVID-19 severity is very important to develop an intensified and personalized treatment for COVID-19 in patients with obesity and/or MS.

Further studies investigating the potential therapeutic effect of specific anti-inflammatories, such as anti-IL-6 antibodies, anti-diabetic drugs or, perhaps, anti-coagulant drugs, in COVID-19 treatment of patients with obesity or MS may be crucial.

This study has potential limitations that should be addressed. The main limitation of the work was the relatively small number of patients, and that there is only one center involved in the study, which may have limited the statistical power of our analysis. Additionally, the treatment carried out for COVID-19 has varied throughout the months of inclusion, according to protocol updates.

## 5. Conclusions

Considering all of our results, the presence of obesity, and especially MS, increases the morbidity and mortality of patients with SARS-CoV-2 pneumonia. Therefore, in these situations, treatment for COVID-19 should likely be intensified and personalized. To monitor the disease, IL-6 can be used as predictive biomarker of fatal outcome in the patients with obesity. Moreover, troponin and LDH are good predictive biomarkers for the oxygen therapy/high-flow nasal cannulas/ MV need in COVID-19 patients with MS. 

## Figures and Tables

**Figure 1 jpm-11-00227-f001:**
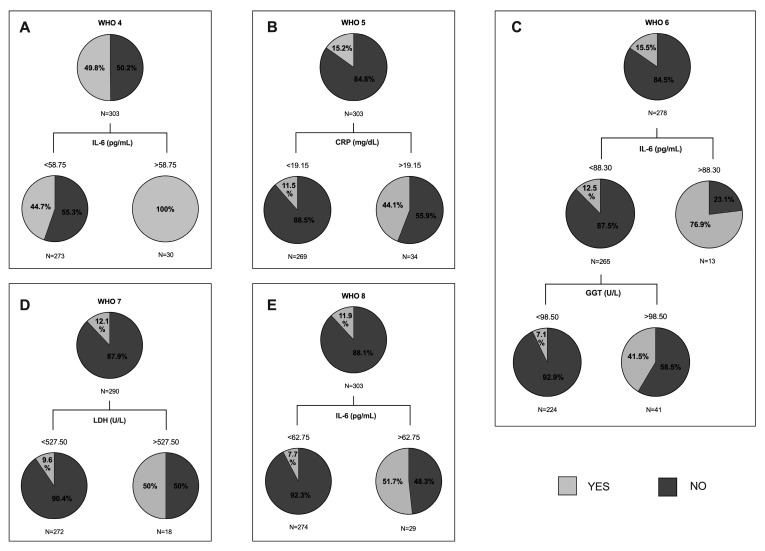
Classification and regression trees of the whole cohort of study were elaborated through CRT method for World Health Organization (WHO) eight-point classification of severe pneumonia by COVID-19 (namely: (0) no clinical or virological evidence of covid-19 infection; (1) infected without limitations; (2) limitation of activity; (3) hospitalized without oxygen therapy; (4) oxygen by mask or nasal; (5) non-invasive ventilation (NIV) (continuous positive airway pressure (CPAP) or positive bipressure in the airways (BiPAP)) or high-flow oxygen (HFO); (6) intubation with mechanical ventilation (MV) or mask with reservoir; (7) MV or extracorporeal membrane oxygenation (ECMO), support with vasopressors, dialysis/renal replacement therapy; and (8) death. (**A**) Pie charts represent the proportion of patients who met the WHO 4 (light grey) or not (dark grey) at each node of the tree. IL-6: interleukin 6. (**B**) Pie charts represent the proportion of patients who met the WHO 5. CRP: c-reactive protein. (**C**) Pie charts represent the proportion of patients who met the WHO 6. IL-6: interleukin 6; GGT: gamma-glutamyl transferase. (**D**) Pie charts represent the proportion of patients who met the WHO 7. LDH: lactate dehydrogenase. (**E**) Pie charts represent the proportion of patients who met the WHO 8. IL-6: interleukin 6.

**Figure 2 jpm-11-00227-f002:**
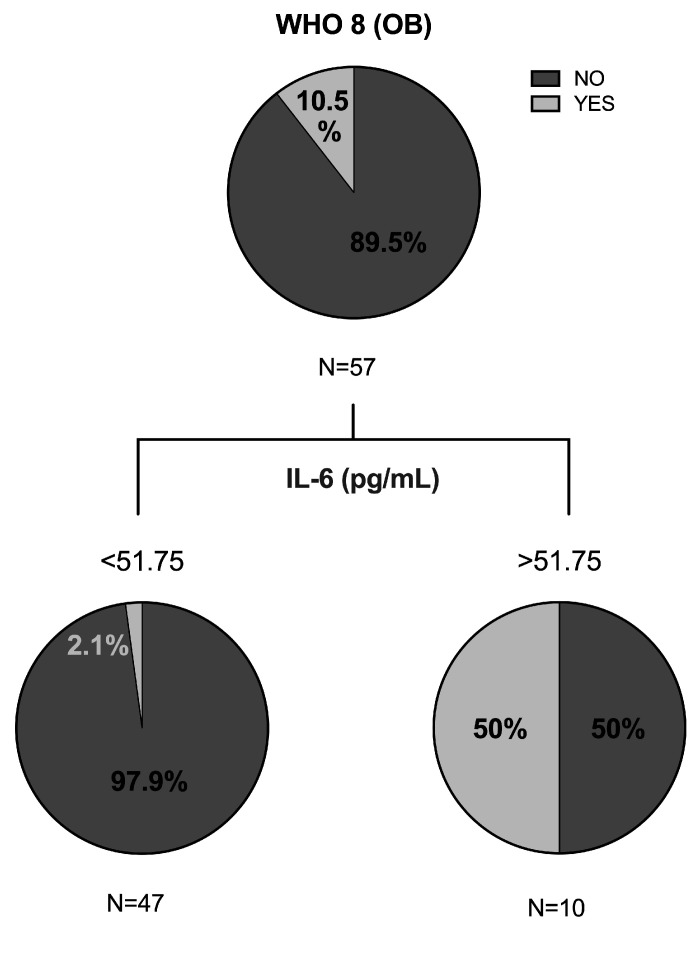
Classification and regression tree of the cohort with obesity were elaborated through CRT method for variable WHO 8 of severe pneumonia by COVID-19, namely: death. OB: obesity; IL-6: interleukin 6.

**Figure 3 jpm-11-00227-f003:**
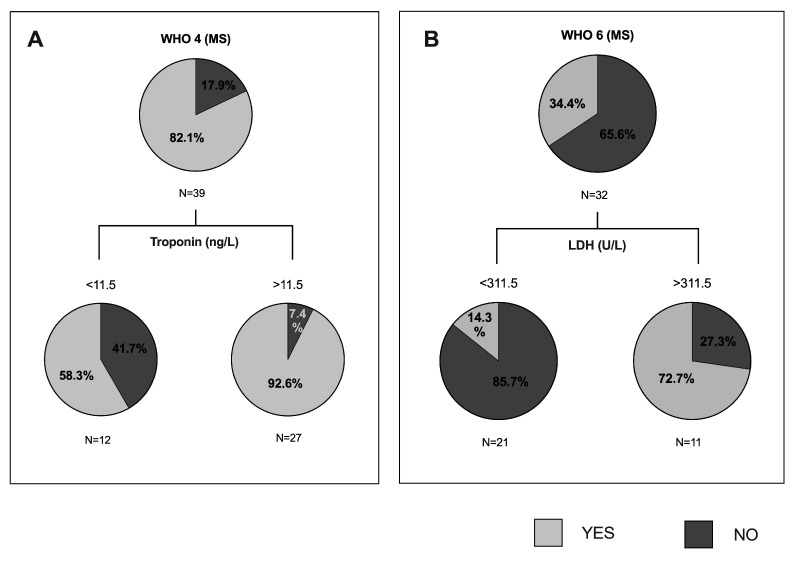
Classification and regression trees of the cohort with metabolic syndrome were elaborated through CRT method for WHO 4 and WHO 6 variables of severe pneumonia by COVID-19, namely: (4) oxygen by mask or nasal; (6) intubation with mechanical ventilation (MV), mask with reservoir. (**A**) Pie charts represent the proportion of patients with MS who met the WHO 4 (light grey) or not (dark grey) at each node of the tree. MS: metabolic syndrome. (**B**) Pie charts represent the proportion of patients who met the WHO 6. MS: metabolic syndrome; LDH: lactate dehydrogenase.

**Table 1 jpm-11-00227-t001:** Background, comorbidities, COVID-19 treatment, and anthropometric, clinical, and biochemical variables of the COVID-19 cohort classified according to the BMI (<30 and ≥30 kg/m^2^).

Variables		WOB (*n* = 176) Mean ± SD; N (%)	OB (*n* = 67) Mean + SD; N (%)	*p*-Value
**Age (years)**		57 (41.25–72)	56 (44–68)	0.678
**Sex (%)**	**Male**	88 (50)	33 (49.3)	0.917
	**Female**	88 (50)	34 (50.7)	
**BMI (kg/m^2^)**		25.1 (22.92–27.34)	34.08 (31.25–37.70)	<0.001 *
**SBP (mmHg)**		126.62 (20.13)	132.58 (20.19)	0.062
**DBP (mmHg)**		77 (69–85)	80 (71–88.25)	0.085
**Background (N (%))**				
**Exercise**		77 (53.5)	17 (30.4)	0.003 *
**Smoking**	**Active**	16 (9.4)	5 (7.7)	0.737
	**Former**	19 (11.2)	7 (10.8)	0.687
**Alcohol consumer**		32 (19.2)	11 (17.5)	0.938
**MS**		10 (5.7)	24 (35.8)	<0.001 *
**T2DM**		20 (11.4)	15 (22.4)	0.029 *
**Dyslipidemia**		51 (29)	20 (29.9)	0.849
**Hypertension**		55 (31.3)	32 (47.8)	0.017 *
**CVD**		30 (17)	8 (11.9)	0.329
**Respiratory diseases**		13 (7.4)	7 (10.4)	0.439
**Cancer**		15 (8.5)	6 (9)	0.915
**Clinical Characteristics (N (%))**				
**Symptoms**	**Mild**	57 (32.4)	12 (17.9)	0.026 *
	**Moderate**	73 (41.5)	39 (58.2)	0.020 *
	**Critical**	36 (20.5)	14 (22.8)	0.940
**ICU admission**		33 (18.8)	13 (20.9)	0.908
**Mortality**		13 (7.4)	8 (11.9)	0.260
**Radiological characteristics**	**Bilateral interstitial pattern**	90 (51.1)	45 (67.2)	0.025 *
	**Pleural effusion**	2 (1.1)	0 (0)	0.382
**Pneumonia**	**Mild**	17 (9.7)	4 (6)	0.361
	**Moderate**	68 (38.6)	37 (55.2)	0.020 *
	**Severe**	37 (21)	17 (25.4)	0.467
**Respiratory failure**		66 (37.5)	34 (50.7)	0.061
**PTE**		4 (2.3)	3 (4.5)	0.359
**Treatment (N (%))**				
**Hydroxychloroquine**		73 (41.5)	30 (44.8)	0.643
**Azithromycin**		63 (35.8)	29 (43.9)	0.246
**L** **opinavir-ritonavir**		59 (33.5)	22 (32.8)	0.919
**Tocilizumab**		3 (1.7)	5 (7.5)	0.025 *
**Interferon**		9 (5.1)	2 (3)	0.477
**Corticosteroids**		44 (25)	23 (34.3)	0.147
**Remdesivir**		21 (11.9)	13 (19.4)	0.134
**Other**		101 (57.4)	47 (70.1)	0.069
**Oxygen Therapy (N (%))**				
**Oxygen mask or nasal**		80 (45.5)	36 (53.7)	0.249
**High-flow nasal cannulas**		27 (15.3)	10 (14.9)	0.936
**NIV**	**CPAP**	3 (1.7)	3 (4.5)	0.214
	**BiPAP**	1 (0.6)	1 (1.5)	0.477
**MV**	**Intubation**	29 (16.5)	9 (13.4)	0.560
	**Mask with reservoir**	9 (5.1)	8 (11.9)	0.063
**MV/ECMO**	**Vasopressors**	22 (12.5)	9 (13.4)	0.846
	**Dialysis**	5 (2.8)	2 (3)	0.312
**Biochemical Parameters** **(Mean (SD); Median (25th–75th Percentiles))**				
**Leukocytes (x10^9^/L)**		6.61 (4.33–3.37)	6.37 (4.97–8.66)	0.727
**Lymphocytes (%)**		15.1 (9–25.4)	16.85 (11–23.02)	0.404
**D-Dimer (ng/mL)**		579 (366–1190)	591 (429–979)	0.559
**ESR (mm)**		64 (38–97.5)	54 (32–73.5)	0.120
**IL-6 (pg/mL)**		8.46 (4.52–29.17)	13.3 (6.7–38.5)	0.174
**Ferritin (ng/mL)**		400 (160.75–831.5)	414 (149.25–633)	0.836
**CRP (mg/dL)**		7.1 (2.6–14)	7.5 (3.35–14.8)	0.568
**Glucose (mg/dL)**		102.5 (86–125)	117 (99–141)	0.006 *
**Total-cholesterol (mg/dL)**		133.2 (37.37)	144.88 (32.30)	0.015 *
**HDL-c (mg/dL)**		31.5 (9.92)	31.25 (8.73)	0.999
**LDL-c (mg/dL)**		70.46 (27.52)	95.96 (27.95)	0.102
**Triglycerides (mg/dL)**		119 (89–224)	142.5 (97.25–156.75)	0.896
**Creatinine (mg/dL)**		0.79 (0.61–0.95)	0.74 (0.66–0.93)	0.714
**AST (U/L)**		31 (23–44)	33.5 (21.75–48)	0.347
**ALT (U/L)**		28 (19–52.75)	32.5 (21.25–58.5)	0.191
**GGT (U/L)**		49 (25–89)	57.5 (38.5–89.75)	0.157
**AP (U/L)**		69 (54–90)	68.5 (54.25–92.5)	0.600
**LDH (U/L)**		263 (221.25–315)	287 (237.5–344)	0.047 *
**Troponin (ng/L)**		8 (3.75–19.25)	6 (3–26)	0.666

WOB: patients without obesity; OB: patients with obesity; N: counts; SD: standard deviation; BMI: body mass index; SBP: systolic blood pressure; DBP: diastolic blood pressure; MS: metabolic syndrome; T2DM: type 2 diabetes mellitus; CVD: cardiovascular disease; ICU: intensive care unit; PTE: pulmonary thromboembolism; NIV: non-invasive ventilation; MV: mechanical ventilation; ECMO: extracorporeal membrane oxygenation; ESR: erythrocyte sedimentation rate; IL-6: interleukin-6; CRP: c-reactive protein; HDL-c: high density lipoprotein-cholesterol; LDL-c: low density lipoprotein-cholesterol; AST: aspartate aminotransferase; ALT: alanine aminotransferase; GGT: gamma-glutamyl transferase; AP: alkaline phosphatase; LDH: lactate dehydrogenase. Quantitative data are expressed as the mean (standard deviation) or median (IQR), depending on the distribution of the variables. Categorical variables are expressed as counts (percentage). * Significant differences between WOB group and OB group (*p* < 0.05).

**Table 2 jpm-11-00227-t002:** Background, comorbidities, COVID-19 treatment, and anthropometric, clinical, and biochemical variables of the COVID-19 cohort with and without MS classified according to the Alberti et al. criteria [20].

Variables				NMS (*n* = 255) Mean ± SD; N (%)	MS (*n* = 48) Mean + SD; N (%)	*p*-Value
**Age (years)**				56 (41–72)	72.5 (64.25–78)	<0.001 *
**Sex (%)**			**Male**	126 (49.4)	33 (68.8)	0.014 *
			**Female**	129 (50.6)	15 (31.3)	
**BMI (kg/m^2^)**				26.06 (23.61–29.31)	31.63 (27.27–35.12)	<0.001 *
**SBP (mmHg)**				127.16 (20.03)	136.56 (18.27)	0.003 *
**DBP (mmHg)**				78 (70.25–86)	75 (70.5–82.5)	0.161
**Background (N (%))**						
**Exercise**				89 (46.6)	5 (15.6)	0.001 *
**Smoking**		**Active**		20 (8.4)	6 (14.3)	0.291
		**Former**		25 (10.5)	11 (26.2)	0.010 *
**Alcohol consumer**				44 (18.9)	10 (25)	0.371
**Obesity**				43 (20.6)	24 (70.6)	<0.001 *
**T2DM**				15 (5.9)	37 (77.1)	<0.001 *
**Dyslipidemia**				57 (22.4)	43 (89.6)	<0.001 *
**Hypertension**				75 (29.4)	48 (100)	<0.001 *
**CVD**				29 (11.4)	23 (47.9)	<0.001 *
**Respiratory diseases**				18 (7.1)	11 (22.9)	0.001 *
**Cancer**				21 (8.2)	6 (12.5)	0.342
**Clinical Characteristics (N (%))**						
**Symptoms**		**Mild**		69 (27.1)	5 (10.4)	0.707
		**Moderate**		124 (48.6)	29 (60.4)	0.014 *
		**Critical**		48 (18.8)	12 (25)	0.135
**ICU admission**				42 (16.5)	9 (18.8)	0.699
**Mortality**				22 (8.6)	14 (29.2)	<0.001 *
**Radiological characteristics**	**Bilateral interstitial pattern**			149 (58.4)	32 (66.7)	0.287
	**Pleural effusion**			3 (1.2)	1 (2.1)	0.451
**Pneumonia**	**Mild**			20 (7.8)	3 (6.3)	0.703
	**Moderate**			117 (45.9)	22 (45.8)	0.995
	**Severe**			54 (21.2)	18 (37.5)	0.015 *
**Respiratory failure**				103 (40.4)	29 (60.4)	0.01 *
**PTE**				6 (2.4)	2 (4.2)	0.473
**Treatment (N (%))**						
**Hydroxychloroquine**				110 (43.1)	20 (41.7)	0.850
**Azithromycin**				103 (40.4)	20 (42.6)	0.782
**L** **opinavir-ritonavir**				82 (32.2)	12 (25)	0.326
**Tocilizumab**				9 (3.5)	4 (8.3)	0.133
**Interferon**				13 (5.1)	1 (2.1)	0.362
**Corticosteroids**				71 (27.8)	17 (35.4)	0.290
**Remdesivir**				38 (14.9)	10 (20.8)	0.303
**Other**				160 (62.7)	38 (79.2)	0.029 *
**Oxygen Therapy (N (%))**						
**Oxygen mask or nasal**				122 (47.8)	30 (62.5)	0.063
**High-flow nasal cannulas**				38 (14.9)	8 (16.7)	0.755
**NIV**			**CPAP**	5 (2)	2 (4.2)	0.351
			**BiPAP**	2 (0.8)	0 (0)	0.539
**MV**			**Intubation**	35 (13.7)	8 (16.7)	0.593
			**Mask with reservoir**	19 (7.5)	6 (12.5)	0.244
**MV/ECMO**			**Vasopressors**	28 (11)	7 (14.6)	0.474
			**Dialysis**	7 (2.7)	1 (2.1)	0.793
**Biochemical Parameters** **(Mean (SD); Median (25th–75th Percentiles))**						
**Leukocytes (x10^9^/L)**				6.62 (4.78–8.70)	75 (70.5)	0.870
**Lymphocytes (%)**				16.8 (9.55–24)	6.47 (4.73–8.55)	0.219
**D-Dimer (ng/mL)**				620 (395.5–1264.5)	764 (435.25–1329.25)	0.295
**ESR (mm)**				60 (36–88.75)	60 (32.25–115.5)	0.745
**IL-6 (pg/mL)**				12.55 (5.22–30.37)	16 (5.88–58.8)	0.349
**Ferritin (ng/mL)**				418 (162–822)	396 (167–556)	0.310
**CRP (mg/dL)**				7.7 (3.1–14)	7.9 (2.45–16.7)	0.725
**Glucose (mg/dL)**				103 (85–124)	132 (107.25–157)	<0.001 *
**Total-cholesterol (mg/dL)**				137.35 (34.45)	132.59 (35.43)	0.572
**HDL-c (mg/dL)**				31.93 (8.94)	25.33 (10.11)	0.284
**LDL-c (mg/dL)**				75.21 (28.72)	60.67 (23.48)	0.529
**Triglycerides (mg/dL)**				121.5 (87.5–221.75)	142.5 (110–165)	0.734
**Creatinine (mg/dL)**				0.78 (0.62–0.96)	0.92 (0.75–1.14)	<0.001 *
**AST (U/L)**				31.5 (24–45)	23 (19–48)	0.098
**ALT (U/L)**				28 (19–52)	28 (16.25–41.75)	0.538
**GGT (U/L)**				49 (25–89)	57.5 (38.5–89.75)	0.157
**AP (U/L)**				69 (54–90)	68.5 (54.25–92.5)	0.600
**LDH (U/L)**				263 (221.25–315)	287 (237.5–344)	0.047 *
**Troponin (ng/L)**				8 (3.75–19.25)	6 (3–26)	0.666

NMS: non-metabolic syndrome patients; MS: metabolic syndrome patients; N: counts; SD: standard deviation; BMI: body mass index; SBP: systolic blood pressure; DBP: diastolic blood pressure; T2DM: type 2 diabetes mellitus; CVD: cardiovascular disease; ICU: intensive care unit; PTE: pulmonary thromboembolism; NIV: non-invasive ventilation; MV: mechanical ventilation; ECMO: extracorporeal membrane oxygenation; ESR: erythrocyte sedimentation rate; IL-6: interleukin-6; CRP: c-reactive protein; HDL-c: high density lipoprotein-cholesterol; LDL-c: low density lipoprotein-cholesterol; AST: aspartate aminotransferase; ALT: alanine aminotransferase; GGT: gamma-glutamyl transferase; AP: alkaline phosphatase; LDH: lactate dehydrogenase. Quantitative data are expressed as the mean (standard deviation) or median (IQR), depending on the distribution of the variables. Categorical variables are expressed as counts (percentage). * Significant differences between NMS group and MS group (*p* < 0.05).

**Table 3 jpm-11-00227-t003:** Significant associations between different variables of COVID-19 patients (anthropometric data, background, treatment, and biochemical variables) and WHO eight-point classification of severe pneumonia by COVID-19 criteria.

Correlations	WHO 4	WHO 5	WHO 6	WHO 7	WHO 8
**Anthropometric Data**					
**Age (years)**	0.336 **	0.182 **	0.19 **	0.093	0.324 **
**BMI (kg/m^2^)**	0.187 **	0.107	0.154 **	0.103	0.067
**SBP (mmHg)**	0.112	−0.017	0.052 *	0.038	0.097
**DBP (mmHg)**	−0.07	−0.094	−0.075	−0.143 *	−0.112
**Background and Comorbidities**					
**Exercise**	−0.308 **	−0.194 **	−0.287 **	−0.22 **	−0.275 **
**MS**	0.107	0.018	0.031	0.041	0.232 **
**T2DM**	0.191 **	0.125 *	0.148 *	0.021	0.212 **
**Dyslipidemia**	0.11	0.075	0.069	0.143 *	0.176 **
**Hypertension**	0.286 **	0.062	0.1	0.091	0.257 **
**CVD**	0.069	−0.046	−0.048	0.002	0.157 **
**Respiratory diseases**	−0.035	0.019	−0.003	−0.082	0.123 *
**Cancer**	0.103	0.126*	0.085	0.063	0.279 **
**COVID-19 Treatment**					
**Hydroxychloroquine**	0.664 **	0.284 **	0.496 **	0.336 **	0.218 **
**Azithromycin**	0.217 **	0.024	0.066	−0.04	0.049
**Lopinavir−ritonavir**	0.569 **	0.293 **	0.462 **	0.372 **	0.107
**Tocilizumab**	0.146 *	0.137 *	0.011	0.044	0.124 *
**Interferon**	0.219 **	0.214 **	0.329 **	0.416 **	0.065
**Remdesivir**	−0.038	0.043	−0.133 *	−0.042	−0.02
**Biochemical Parameters**					
**Leukocytes (x10^9^/L)**	0.057	0.024	0.16 *	0.101	0.087
**Lymphocytes (%)**	−0.221 **	−0.217 **	−0.277 **	−0.204 **	−0.158 *
**D-Dimer (ng/mL)**	0.246 **	0.119	0.192 *	0.166 **	0.236 **
**ESR (mm)**	0.19 *	0.097	0.114	0.138	0.036
**IL-6 (pg/mL)**	0.285 **	0.114	0.32 **	0.222 **	0.262 **
**Ferritin (ng/mL)**	0.194 *	0.162 *	0.204 **	0.15 *	0.082
**CRP (mg/dL)**	0.334 **	0.223 **	0.323 **	0.177 **	0.085
**Glucose (mg/dL)**	0.129 *	0.038	0.116	0.106	0.165 **
**Triglycerides (mg/dL)**	0.501 **	0.122	0.289	0.188	0.074
**Creatinine (mg/dL)**	0.056	−0.026	0.065	0.091	0.212 **
**AST (U/L)**	0.104	0.106	0.137 *	0.111	0.02
**GGT (U/L)**	0.243 **	0.121	0.15 *	0.186 **	−0.054
**AP (U/L)**	0.092	0.021	0.146 *	0.117	0.09
**LDH (U/L)**	0.299 **	0.22 **	0.286 **	0.21 **	0.15 *
**Troponin (ng/L)**	0.281 **	0.124	0.27 **	0.244 **	0.319 **

BMI: body mass index; SBP: systolic blood pressure; DBP: diastolic blood pressure; MS: metabolic syndrome; T2DM: type 2 diabetes mellitus; CVD: cardiovascular disease; ESR: erythrocyte sedimentation rate; IL-6: interleukin-6; CRP: c-reactive protein; AST: aspartate aminotransferase; GGT: gamma-glutamyl transferase; AP: alkaline phosphatase; LDH: lactate dehydrogenase; N: counts; WHO 4: oxygen by mask or nasal; WHO 5: non-invasive ventilation (continuous positive airway pressure [CPAP] or positive bipressure in the airways [BiPAP]) or high-flow oxygen; WHO 6: intubation with mechanical ventilation, mask with reservoir; WHO 7: mechanical ventilation or extracorporeal membrane oxygenation, support with vasopressors, or dialysis/renal replacement therapy. All correlations were expressed as Spearman rho correlation coefficient; WHO 8, death. * Significant association with a *p*-value < 0.05. ** Significant association with a *p*-value < 0.01.

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
