# Peer review of "Predictive Biomarkers of COVID-19 Severity in SARS-CoV-2 Infected Patients with Obesity and Metabolic Syndrome"

_jpm, 2021, doi:10.3390/jpm11030227_

Round 1

Reviewer 1 Report

The authors described a relevant topic, especially for the current period. They studied whether certain variables whether prognostic biomarkers described in patients with COVID-19 were capable to predict the evolution of disease in patients with this infection who have obesity or metabolic syndrome. The manuscript was well written. The main limitations of the paper were the small number of patients and it was a single center study. However, here my comments/recommendations:

1. Line 61: Obesity and metabolic dysregulation are also an underlying cause of fibrosis, including idiopathic pulmonary fibrosis. The role of metabolic dysregulation in the pathogenesis of IPF has not been extensively studied, despite a recent surge of interest such as the following paper:

  • Zank DC, Bueno M, Mora AL, Rojas M. Idiopathic Pulmonary Fibrosis: Aging, Mitochondrial Dysfunction, and Cellular Bioenergetics. Front Med (Lausanne). 2018 Feb 5;5:10. doi: 10.3389/fmed.2018.00010. PMID: 29459894; PMCID: PMC5807592.

On this topic, very recently, few papers investigated the role of adipokines in different lung diseases. Adipokines are mainly secreted by adipocytes, macrophages and various other cells, along with their role in the regulation and mediation of inflammatory responses, as reported in the following papers:

  • Enomoto N, Oyama Y, Yasui H, Karayama M, Hozumi H, Suzuki Y, Kono M, Furuhashi K, Fujisawa T, Inui N, Nakamura Y, Suda T. Analysis of serum adiponectin and leptin in patients with acute exacerbation of idiopathic pulmonary fibrosis. Sci Rep. 2019 Jul 19;9(1):10484. doi: 10.1038/s41598-019-46990-3. PMID: 31324858; PMCID: PMC6642131.
  • d'Alessandro M, Bergantini L, Cameli P, Lanzarone N, Perillo F, Perrone A, Bargagli E. BAL and serum multiplex lipid profiling in idiopathic pulmonary fibrosis and fibrotic hypersensitivity pneumonitis. Life Sci. 2020 Sep 1;256:117995. doi: 10.1016/j.lfs.2020.117995. Epub 2020 Jun 20. PMID: 32574666.

2. Line 72: The authors stated "Subjects with obesity and T2DM have disrupted innate and adaptive immune responses" Please provide references such as:

  • Benetti E, Giliberti A, Emiliozzi A, Valentino F, Bergantini L, Fallerini C, Anedda F, Amitrano S, Conticini E, Tita R, d'Alessandro M, Fava F, Marcantonio S, Baldassarri M, Bruttini M, Mazzei MA, Montagnani F, Mandalà M, Bargagli E, Furini S; GEN-COVID Multicenter Study, Renieri A, Mari F. Clinical and molecular characterization of COVID-19 hospitalized patients. PLoS One. 2020 Nov 18;15(11):e0242534. doi: 10.1371/journal.pone.0242534. PMID: 33206719; PMCID: PMC7673557.

3. Materials, methods, populations and statistical analysis were correctly described. However, Line 87, Please report the total number of enrolled patients.

4. Line 104: The authors stated "On the other hand, COVID-19 patients were classified depending on the presence of MS (MS, n=48; non-MS, n=255)" as reported in table 2, while in table 1 they reported WOB n=176 and OB n=67. In table 3 they reported 350 patients. It's difficult to understand how many patients were enrolled in the present study and how many patients were used for statistical analysis. Please, check the results.

5. Line 322: the authors stated "In contrast, the practice of physical activities seemed to strengthen the immune system, suggesting a benefit in the COVID-19 clinical course, which agrees with our results that showed a negative association between the practice of physical exercise and all WHO severity criteria analysed (WHO 4 to WHO 8)". However, the sentence is bit confusing, please rephrase.

Reviewer 2 Report

This mauscript ''Predictive biomarkers of COVID-19 severity in SARS-CoV-2 infected patients with obesity and metabolic syndrome'' by Carles Perpiñan et al. Authors investigate the impact of obesity and metabolic syndrome (MS) on the clinical course of COVID-19 and whether prognostic biomarkers described are useful to predict the evolution of COVID-19 in patients with obesity or MS. This manuscript is well write and clear. minor comments: 1. Typo: line 167, with obesity BMI should ≥30 kg/m2 2. Is there any different between male and female patients with OB and MS after infected COVID-19? How about young and senior patients under the similar condition?

Reviewer 3 Report

The authors present solid data and useful results. I do appreciate the work done and recommend considering the following issues:

  1. At the end of the Abstract, the main results need to be presented more clearly and explicitly.
  2. Tables and Figures should be better described, and the main results need be presented.
  3. In the Discussion, it is not clear how the authors came to the following conclusions:

Lines 338-341:

»Biochemical parameters, such as lymphocyte count, D-dimer, IL-6, CRP, GGT, LDH and troponin, that used to be associated with a hyperinflammatory response had a positive association with worse prognosis (different WHO severity criteria for COVID-19 pneumonia) in our study, which was previously reported [31,32].«

Lines 343-344:

»We found that the best predictor of WHO scores of 4, 6 and 8 was IL-6. Other studies have also described IL-6 as a predictor of mortality associated with...«

The conclusions need to be linked directly with the results obtained in the manuscript. Make reference to the results presented in Tables and Figures.  

Minor Comment: Please check all references; there are missing names of authors and missing DIOs.

Round 2

Reviewer 1 Report

The authors replied exhaustively to the comments.